# Segmental External Load in Linear Running in Elite Futsal Players: A Multifactorial and Individual Variability Analysis Using Linear Mixed Models

**DOI:** 10.3390/sports13080268

**Published:** 2025-08-13

**Authors:** Diego Hernán Villarejo-García, Carlos Navarro-Martínez, José Pino-Ortega

**Affiliations:** Faculty of Sport Science, University of Murcia, 30100 Murcia, Spain; dvillarejo@um.es (D.H.V.-G.); josepinoortega@um.es (J.P.-O.)

**Keywords:** external load monitoring, female athletes, futsal, inertial measurement units, linear running, load distribution, segmental PlayerLoad, training load

## Abstract

Limited evidence exists on how segmental external load is distributed during linear running and how it varies with speed, training intensity, and individual differences. This study examines the external load profile across six body segments in elite female futsal players during linear treadmill running, focusing on the effects of speed and training zone, as well as individual variability. Eight elite players, including six outfield players and two goalkeepers (mean age 23.9 ± 3.4 years, height 164.96 ± 4.22 cm, body mass 60.31 ± 4.56 kg), performed an incremental test and were measured using six WIMU PRO™ inertial sensors. The sensors recorded segmental PlayerLoad, speed, and training zones. Data were analyzed using Linear Mixed Models. The most important results show significant interactions between body location and speed and between body location and training zone (*p* < 0.001), with intraclass correlation coefficients (ICC) ranging from 0.437 to 0.515. These results indicate variability among players and specific and asymmetrical segmental load patterns. These findings offer practical insights for tailoring individualized training strategies that optimize performance and reduce segment specific overuse injuries.

## 1. Introduction

The monitoring of segmental external load in elite training is a key factor for achieving higher performance in athletes [1]. This individualization allows us to optimize athletic performance [2] and reduce the risk of injury [3]. In this way, external load [4] and, in particular, metrics such as PlayerLoad [5], which can be obtained using various inertial sensors, can provide valuable information regarding the mechanical demands placed on different body segments of high-performance athletes during their daily training. Specifically, one of the sports with the most intense physical demands is futsal. In this sport, players are exposed to very high heart rate intensities [6], frequent and numerous changes of direction [7], and intermittent effort throughout the match [8].

Physical preparation in female futsal players uses various training methodologies aimed at enhancing or replicating specific aspects of the game. Among the most common training methods are small-sided games (SSGs) [9], running drills involving changes of direction (COD) [10], and linear running [11]. Small-sided games (SSGs) are known for providing high contextual and cognitive specificity, although managing the training load in these activities tends to be more variable and challenging to measure [12]. Meanwhile, running drills with changes of direction (COD) are designed to replicate the multidirectional movement patterns typical of futsal, although they may lack the tactical complexity found in actual match play. Conversely, linear running, while less specific in terms of multidirectionality and the inherent intermittency of futsal, remains a prevalent training method due to its ease of prescription, precise speed control, and intensity monitoring.

In this regard, although running speed acts as a modulator of the total load imposed on the athlete, both in terms of internal load (e.g., heart rate) and external load (e.g., distance covered), a crucial question arises in the context of linear treadmill running: Does external load, measured segmentally, behave the same way across all body locations when speed varies? More importantly, is this segmental load profile consistent among all players, or does significant individual variability exist? In this respect, Gómez-Carmona et al. examined segmental external load profiles in young football players during an incremental treadmill test, providing an initial insight into how load is distributed across different body locations [13]. Their results show that PlayerLoad differs significantly between various body segments in response to linear running. However, the statistical analysis used in that study, based on Analysis of Variance (ANOVA), did not allow for modeling of the complex interactions between factors, nor the inherent variability of repeated measures within the same individual, which could limit the depth of interpretation of their findings. Therefore, there remains a gap in the detailed understanding of how speed and internal load (training zone) specifically modulate external load at the segmental level in elite female players, and how this manifests in inter-individual variability during linear running. Unlike previous studies, such as Gómez-Carmona et al. [13], which investigated external load in male futsal players and employed Analysis of Variance (ANOVA) to compare loads between different positions or game conditions without addressing individual variability or complex interactions between multiple factors, the present study utilizes Linear Mixed Models. This advanced statistical approach allows for a more robust analysis of multifactorial effects and individual differences, providing unprecedented insight into segmental load patterns. Furthe]. Understanding on elite female futsal players addresses a critical research gap, as this population experiences high physical demands and potential injury vulnerabilities, yet remains underrepresented in load monitoring research [14,15]. Understanding their specific load profiles is crucial for optimizing performance and injury prevention strategies tailored to their unique physiological and biomechanical characteristics.

Given that linear running is a widely used training method, addressing this knowledge gap is essential for optimizing the prescription of linear running training, enabling coaches to better understand the specific demands placed on each body segment and thereby contributing to performance enhancement and injury prevention. The application of a Linear Mixed Model (LMM) is key for this purpose, as it enables the analysis of complex interactions between factors and allows for modeling of the significant variability inherent in individual athlete data [16,17].

Accordingly, the present study aims to address this gap by investigating external load in elite female futsal players from a multifactorial and individual-centered perspective. Thus, the objectives of this work were as follows: (1) to characterize the differential influence of speed on segmental external load according to body location; (2) to determine specific patterns of external load by body location across different training intensity zones; and (3) to quantify the individual variability in external load attributable to differences between players. It was hypothesized that segmental external load (measured using the PlayerLoad index) would be influenced by speed and training zone and that these effects would vary substantially depending on body location, with considerable variability also expected among players. The results of this study offer valuable practical implications for coaches and strength and conditioning professionals, as they enable the individualization of linear running training, optimization of load management strategies, and contribute to the prevention of segment-specific injuries in elite female futsal players.

## 2. Materials and Methods

### 2.1. Participants

The study sample comprised 8 elite-level female futsal players (6 outfield players, 2 goalkeepers) (age: 29.9 ± 5.1 years; height: 164.96 ± 4.22 cm; body mass: 60.31 ± 4.56 kg) who participated in the Primera RFEF Futsal Femenina (Spanish first division) during the 2020/2021 season. The present study adopted an observational within-subject repeated measures design. All players were professional athletes with at least five years of experience in elite futsal and presented no musculoskeletal injuries or physical limitations that could have affected their performance during the study.

Following the Declaration of Helsinki guidelines, and to ensure participant confidentiality, all performance data were anonymized prior to analysis [18]. The study protocol was approved by the Ethics Committee of the University of Murcia (approval number: 3180/2020), which waived the requirement for written informed consent, as the research was deemed non-interventional, utilizing retrospective and anonymized routine performance data collected within a professional sports team setting. The study sample comprised 8 elite-level female futsal players… All players were actively participating in regular training with their professional team without current or recent (within 6 months) history of lower-limb or spinal injuries requiring medical intervention or preventing full participation.

### 2.2. Equipment

For external workload assessment during the incremental running treadmill test, six WIMU PRO™ inertial devices (Realtrack Systems, Almería, Spain) were utilized. These devices are equipped with triaxial accelerometers capable of detecting and measuring movement through a micro-electromechanical system. In the present research, the accelerometers’ sampling frequency was set at 100 Hz. Data acquisition and initial processing were performed using the SPRO™ software (version 989). A motorized treadmill (HP Cosmos Pulsar 3P, HP Cosmos Sports & Medical GmbH, Nussdorf-Traunstein, Germany), with a minimum capacity of 20 km/h and electronic speed control, was also used for testing. To ensure proper sensor attachment, a back harness, elastic leg straps, and industrial-grade Velcro were employed.

### 2.3. Variables

#### 2.3.1. Dependent Variable

PlayerLoad: PlayerLoad can be defined as a measurement that indicates the accumulated external load that an athlete—in this, case professional female futsal players—has to endure. This measurement is calculated based on acceleration data recorded in three directions (forward–backward, side-to-side, and up–down). The final value is obtained by summing all the high-speed changes in acceleration recorded across these three axes. This indicator was developed by Realtrack Systems (Almería, Spain), and is expressed in arbitrary units (a.u.). It is consolidated into a single value, resulting from the sum of the differences in acceleration between each moment across the three directions.$$\mathrm{PlayerLoad}=\sum \surd ({\left(A_{\mathrm{xi}}-A_{\mathrm{xi}-1}\right)}^{2}+{\left(A_{\mathrm{\gamma i}}-A_{\mathrm{\gamma i}-1}\right)}^{2}+{\left({Az}_{i}-{Az}_{i-1}\right)}^{2})$$

In this formula, A_xi_, A_yi_, and A_zi_ denote the acceleration values on the X, Y, and Z axes at time point i, and *n* is the total number of samples. This parameter reflects the magnitude of instantaneous mechanical load fluctuations during movement, and has been widely used to monitor external workload in team sports [19,20].

#### 2.3.2. Independent Variables

Speed: This is defined as the distance covered in a given period of time and is commonly expressed in kilometers per hour (km/h).

Training Zone: This is defined as the classification of exercise intensity based on the estimated percentage of an individual’s maximal heart rate [21]. Five distinct training zones were defined for analysis: R0 (50–60% HRmax), R1 (60–70% HRmax), R2 (70–80% HRmax), R3 (80–90% HRmax), and R4 (>90% HRmax).

Location: This is defined as the anatomical placement of the inertial measurement units (IMUs) during the exercise, allowing for the recording of load data in different body regions (see Figure 1). Once the data were obtained using the SPRO™ software, they were processed in Microsoft Excel 365.

### 2.4. Statistical Analysis

The PlayerLoad variable was subjected to a natural logarithm transformation prior to the analysis. This transformation was performed primarily to address the positive skewness in the distribution of external load and to improve the normality and homoscedasticity of the model residuals [17]. To manage the high collinearity between speed and training zone, two separate Linear Mixed Model (LMM) analyses were employed. Both models included ID_jugadora as a random intercept to model individual variability among players. Additionally, Model 1 included speed as a random slope to account for individual variability in the response to speed. Model 1 investigated the effects of location, speed, and their two-way interaction (Location × Speed). Model 2 investigated the effects of location, training zones, and their two-way interaction (Location × Training Zone). Prior to LMM execution, statistical assumptions were verified for each model. The normality of the model’s residuals was evaluated using the Kolmogorov–Smirnov test, complemented by visual inspection of Q-Q plots. For Model 1, the K-S test yielded *p* < 0.001; however, visual inspection of the Q-Q plot indicated a close approximation to normality, particularly in the central distribution, suggesting that the transformation substantially mitigated non-normality for practical inference, given the large sample size. For Model 2, the K-S test yielded *p* = 0.093; the Q-Q plot showed an excellent approximation to normality. The homoscedasticity of residuals and the linearity of the relationship between predictor variables and the response variable were visually inspected using scatter plots of the residuals versus predicted values, with no significant deviations observed in either model. Statistical significance for all fixed effects was set at *p* < 0.05. For a more complete interpretation of the practical magnitude of the effects, effect sizes were calculated using Partial Eta Squared (η*p*^2^). These values were interpreted, where 0.01 is considered a small effect, 0.06 a medium effect, and 0.14 a large effect [22]. For a detailed interpretation of significant interactions, conceptual simple effects analyses were performed based on parameter estimates and confidence intervals. All statistical analyses were carried out using Jamovi, version 2.6 [23], and its modules, along with supplementary computations in R Studio, version 4.5.0 [24], to derive the effect sizes (η*p*^2^).

### 2.5. Data Availability

The data used for statistical analysis and the results of this work are publicly available in the Research Data Repository (DIGITUM) under the following Digital Object Identifier (DOI): http://hdl.handle.net/10201/156980 (accessed on 1 July 2025).

### 2.6. Procedure

The research was conducted over three weeks, with one session per week. In the first week, participants were informed about the study’s objectives and protocol. In the second week, they familiarized themselves with the treadmill test procedure and the experimental equipment (inertial sensors) by conducting a trial session. The incremental treadmill test was performed in the third week.

Prior to the placement of the inertial devices, they were manually calibrated according to the manufacturer’s recommendations and synchronized. This process is fundamental for eliminating 3D accelerometer error sources (offset error, scaling error, non-orthogonal error, and random error). The devices were affixed to the following anatomical locations following a validated scheme: (i) upper back, on the interscapular line; (ii) lumbar region, at L3 near the center of mass; (iii) knee, 3 cm above the kneecap; and (iv) ankle, 3 cm above the lateral malleolus. At both the knee and ankle, the devices were placed on the outside of the right leg on all athletes, while an anatomically designed harness was used for the upper back, and the rest were secured with a specifically designed elastic band. Before the test, participants performed a standardized 10 min warm-up that included 5 min of continuous running at 7 km/h, 3 min of joint mobility exercises, and 2 min of progressions (gentle accelerations on the treadmill). This procedure was monitored in real-time via WIMU PRO™ devices (RealTrack Systems, Almería, Spain) that transmitted data wirelessly to the SPRO™ software, ensuring the correct functioning of the sensors. The main protocol was an adaptation of the VAM (Maximal Aerobic Speed) test, designed for continuous and controlled analysis. The test began at a speed of 8.0 km/h, with increments of +0.2 km/h every 12 s. The test concluded upon the athlete’s voluntary exhaustion or observation of a technical criterion (e.g., unstable stride frequency, evident fatigue). The treadmill surface was flat (1% incline). Upon completion of the main test, participants performed 5 min of recovery intensity running. To minimize interference from uncontrolled variables, all participants were instructed to maintain their habitual lifestyle and normal dietary intake before and during the study. Furthermore, tests were always performed at the same time of day (i.e., 9:00 a.m.) to avoid the possible effects of circadian rhythms, and it was ensured that they had not performed high-intensity physical activity in the 48 h prior to the tests. Environmental control was kept constant (temperature 22–24 °C, humidity < 50%). All participants used standard training footwear to homogenize conditions. The integrity of the sensors was verified at all times by two researchers.

## 3. Results

The results of the fixed effects omnibus tests of the Linear Mixed Model are presented in Table 1.

As can be observed in Table 1, statistically significant effects were found for location (Model 1: F5.1607 = 3391.9 *p* < 0.001, η*p*^2^ = 0.914; Model 2: F5.1589 = 497.5 *p* < 0.001, η*p*^2^ = 0.610), Speed (Model 1: F1.1608 = 455.5 *p* < 0.001, η*p*^2^ = 0.739), and training zone (Model 2: F4.1590 = 744.2 *p* < 0.001, η*p*^2^ = 0.652). All these effects showed a very large effect size. Additionally, significant interactions were found between location and speed (Model 1: F5.1607 = 88.6 *p* < 0.001, η*p*^2^ = 0.216), as well as between location and training zone (Model 2: F20.1589 = 11.6 *p* < 0.001, η*p*^2^ = 0.127). Both interactions revealed a large effect size.

Regarding individual variability, the random components of the models revealed that, in Model 1 (which included speed), the variance of the random intercept for player ID (ID_jugadora) was 0.0185 (SD = 0.136), resulting in an ICC of 0.437. This value suggests that 43.7% of the total variability in PlayerLoad (log-transformed) can be attributed to consistent differences among players in this model, and the unexplained residual variance was 0.0239 (SD = 0.154). For Model 2 (which included the training zone), the variance of the random intercept was 0.0348 (SD = 0.186) with an ICC of 0.515, indicating that 51.5% of the total variability in PlayerLoad (log-transformed) is attributable to consistent differences among players. The unexplained residual variance for Model 2 was 0.0327 (SD = 0.181).

Table 2 presents the parameter estimates for the fixed effects of Model 1.

The results in Table 2 show how segmental external load varies significantly with body location and running speed. The right ankle (RA) showed a significantly higher PlayerLoad than the left ankle (LA) (estimate = 0.028, *p* = 0.045), representing an approximate 2.8% increase in load. Likewise, each 1 km/h increment in speed was associated with an approximate 12.5% increase in the PlayerLoad of the left ankle (estimate = 0.117, *p* < 0.001).

Significant location × speed interactions (*p* < 0.05 for all) indicate that the percentage rate of PlayerLoad increase with speed differs among segments.

Table 3 presents the parameter estimates for the fixed effects of Model 2.

The results in Table 3 show that PlayerLoad varies significantly among segments and increases with the intensity of the training zone (R4–R0: estimate = 0.992, *p* < 0.001; 170.9% increase). A significantly higher PlayerLoad was also observed in RA compared to LA (estimate = 0.107, *p* < 0.001; 11.3% increase in R0). Significant location × training zone interactions (*p* < 0.05) reveal that the percentage increment of PlayerLoad with intensity is attenuated in RA, LS, and TS compared to LA.

Figure 2 presents the fixed effects profiles for PlayerLoad as a function of the combinations of location, speed, and training zone.

As observed in Figure 2a, the relationship between PlayerLoad and speed shows a non-uniform dynamic among segments, with more pronounced percentage increases in the ankles. Similarly, Figure 2b shows how the effect of the training zone on PlayerLoad differs according to location.

## 4. Discussion

The aim of this study was to describe and analyze the segmental external load profile in elite futsal players. An observational within-subject repeated measures model was used. Specifically, the influence of running speed, training zone, and inter-individual variability were analyzed. The findings confirm that segmental external load, measured using the PlayerLoad index [25], is significantly influenced by speed and training intensity. Individual variability in load response has also been observed. This confirms the general hypothesis of this work, and provides key information to optimize training and to individualize the effects of linear running in the observed sample.

To analyze the obtained data, two separate Linear Mixed Models (LMMs) were employed. This multivariate approach allowed for addressing the hierarchical structure of repeated measurements performed on the same futsal players [26]. The decision to use two separate LMMs allowed for controlling the high collinearity between speed and training zone. This enabled a reliable estimation of their independent effects and interactions [27]. LMMs also allowed for modeling individual variability among the athletes and understanding the interactions between body location, speed, and training zone [17].

Regarding the analysis of the results, specifically the omnibus tests for fixed effects (Table 1), it was observed that both the main effects of location, speed (Km/h) (in Model 1), training zone (in Model 2), and their interactions were highly significant (*p* < 0.001 in all cases). These findings confirm the interdependence between segmental load and speed and intensity factors. In this regard, the effect sizes (η*p*^2^) also revealed that location, speed, and training zone showed very large effects (ranging between η*p*^2^ = 0.610 and 0.914), and the Location × Speed and Location × Training Zone interactions showed large effects (η*p*^2^ = 0.216 and 0.127, respectively).

Regarding individual variability, the intraclass correlation coefficients were 0.437 for Model 1 and 0.515 for Model 2. These values show that variability in PlayerLoad is attributable to differences among players [26].

This difference in segmental load suggests that, although running speed increases, biomechanical load is not uniformly distributed across the analyzed body segments. This indicates the complexity of the process by which the body adapts to force absorption and transmission during linear running in the analyzed futsal players.

The results of Model 1, regarding the Location × Speed interaction, indicate that distal areas such as the ankles (LA, RA) exhibit a more pronounced percentage increase in PlayerLoad as speed increases. This may be due to the fact that, during the ground contact phase of running, the ankles can be one of the first joints to absorb this impact [28]. As speed increases, ground reaction forces (GRFs) acting on the body increase considerably, leading to more direct and proportional changes in acceleration and, consequently, in the PlayerLoad recorded in these joints [25].

In contrast, the knees (LK, RK) and the upper and lower trunk (LS, TS) showed a more attenuated percentage increase in PlayerLoad with increasing speed, as revealed by the interaction parameters of Model 1. This may be due to their greater ability to reduce, distribute, and balance the forces produced during running [29]. For example, the knees can dynamically adjust their flexion and stiffness to better absorb impact and generate force in later phases of running compared to the ankles, which translates into less marked changes in acceleration and, consequently, in PlayerLoad, even at higher running speeds [30].

Similarly, in the trunk (LS and TS), PlayerLoad shows a smaller percentage increase at high speeds. This may be due to a biomechanical adaptation where the trunk tends to become more rigid and stable to maintain balance during linear running. This translates into a reduction in movement and, therefore, a smaller and less proportional increase in PlayerLoad in these regions [31].

These findings on the variation in segmental load due to speed are consistent with Gómez-Carmona et al. [13], who also reported that the dynamics of accelerometric load differed across body locations and segments at various running speeds in young football players. Nevertheless, the use of Linear Mixed Models in the present study, specified in two separate models for greater precision, allowed for understanding these interactions, revealing percentage change patterns not previously described in the literature.

Along these lines, the results found in the location × training zone interaction are also relevant (Model 2: F(20,1589) = 11.6, *p* < 0.001, η*p*^2^ = 0.127). With a large effect size, the results of this study show that the influence of the training zone on segmental PlayerLoad differs significantly according to body location. This also indicates specific load patterns for each intensity level.

The parameters of Model 2 detail how the PlayerLoad observed in the LA, as the reference location, progressively increases with the intensity of the training zone. For example, an increase of 24.6% from R0 to R1 was observed, reaching an increase of 170.9% from R0 to R4. However, the significant location × training zone interactions (Table 3, Figure 2b) revealed that this increasing percentage pattern differs notably for other locations.

When analyzing the significant interactions of Model 2, an atypical pattern is observed in RA in Figure 2b, as PlayerLoad decreases from zone R0 to R1. This atypical pattern can be explained by the presence of asymmetries in the limbs of Futsal players, related to the dominance of one leg [32], or with neuromuscular changes occurring when transitioning from R0 to R1 [33]. The singularity of this pattern and its potential implications for optimizing performance and preventing injuries should be addressed in new lines of research.

Unlike previous studies that analyzed load in different body parts and at various speeds [13,34], the present study is the first to describe in detail the complex relationship between body location and training zones, as defined by maximal heart rate. Our findings confirm that segmental PlayerLoad varies as a function of effort intensity, an aspect scarcely explored in the literature within the context of linear running using multimodal measurements. These results are particularly useful for planning high-intensity training loads in elite female futsal players, allowing for more precise individualization based on real segmental demands.

Another important point of this study is the individual variability found in segmental PlayerLoad. The observed intraclass correlation coefficients (ICCs), which ranged between 0.437 (Model 1) and 0.515 (Model 2), indicate that approximately between 43.7% and 51.5% of the total variation in PlayerLoad is due to differences among professional players, rather than random factors or measurement error. This demonstrates that, even in a controlled and standardized task such as linear running, the external load response is not identical for all athletes. These findings are consistent with previous studies that examined how factors such as age [32], sex [33], injuries [35], or physical condition [36] influence linear running mechanics, highlighting the need to consider the biomechanical and physiological particularities of each player.

This existing difference among the players in the present study generates valuable practical implications for designing linear running training. In this way, these findings indicate that group-level planning without individualization could even be harmful for some athletes. As observed in the sample of this study, the response to external load in linear running is highly individualized and could be related to differences in running techniques, biomechanics, body composition, or the individual neuromuscular efficiency of each athlete. Therefore, it is essential to carry out continuous and personalized monitoring of external load at the segmental level to ensure that each athlete receives training tailored to their individual needs and characteristics.

## 5. Practical Implications and Future Applications

A practical application of this knowledge is that, by understanding that certain body segments present asymmetrical load patterns or differential responses to intensity and speed (such as the higher basal load and the attenuated increase pattern in the right ankle, or the smaller percentage increases in the trunk at high intensities), coaches can adjust training loads or include specific exercises to compensate for these particular demands in concrete segments. These data can also aid in the design of sports equipment. In this sense, it is necessary to utilize more ergonomic equipment adapted to the anatomical and biomechanical demands of each body segment. In this regard, IMU manufacturers should consider designing miniaturized devices that allow for the simultaneous use of multiple sensors on different body parts, or that can even be integrated directly into sportswear. Similarly, conducting individualized and continuous monitoring of external load over time becomes key to optimizing performance and preventing injuries in specific body areas, thus managing load according to each athlete’s intrinsic needs.

## 6. Study Limitations and Future Research Directions

The present study presents several limitations that should be considered. One of these limitations is that data collection was carried out on a treadmill during linear running. While this controlled environment allowed for the precise manipulation and isolation of variables such as speed and intensity, it does not fully replicate the dynamic reality of futsal. Futsal involves frequent multidirectional movements, highly intermittent efforts, and constant decision-making processes in varied game situations, such as cutting, pivoting, and acceleration/deceleration with and without the ball. This can contribute differently to segmental load, thus affecting the direct transferability of these specific load profiles to real-game situations. Nevertheless, this standardized approach was considered necessary to investigate in isolation the specific effects of speed and intensity on segmental load. Another limitation is the sample size. Although this is common practice in studies with high-performance athletes, it implies that the results should be considered within this field and should not be generalized to broader populations. This limited sample size also restricts the complexity of random-effects structures that could be robustly estimated within the Linear Mixed Models, influencing aspects such as the individual variability of random slopes. Furthermore, while the logarithmic transformation substantially improved the normality of the residuals, the formal statistical tests for normality remained significant for Model 1, suggesting minor deviations from perfect normality; however, visual inspection of Q-Q plots indicated practical acceptability, supported by the known robustness of LMMs with large sample sizes.

For future research, it would be important to confirm these segmental load profiles and interaction patterns in competition and training situations. Specifically, it would be beneficial to deepen the understanding of atypical load patterns observed in the right ankle and differential load responses in trunk segments across intensities. Furthermore, it is necessary to replicate this research in other populations. Future studies could also benefit from the analysis of other variables, such as perceived internal load, individual players’ playing profiles, and limb dominance.

## 7. Conclusions

The present study allows us to describe how external load affects different body parts in professional female futsal players during linear treadmill running. The results show a differentiated impact of running speed and training zones on distinct body segments. Specific and asymmetrical segmental load patterns were observed, alongside considerable individual variability among athletes. These findings are important for understanding the significance of monitoring load in different body segments. They also highlight the need to adjust training loads by considering each player’s demands. This approach is key to optimizing performance and contributing to injury prevention in this elite population.

## Figures and Tables

**Figure 1 sports-13-00268-f001:**
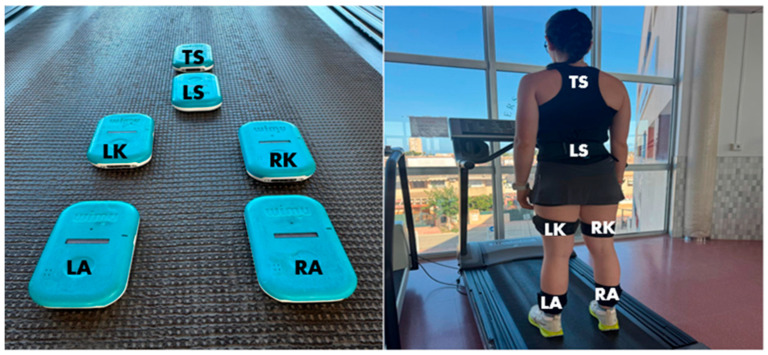
Anatomical placement of inertial measurement units (IMUs). Left ankle (LA), right ankle (RA), left knee (LK), right knee (RK), lumbar spine (LS), and thoracic spine (TS).

**Figure 2 sports-13-00268-f002:**
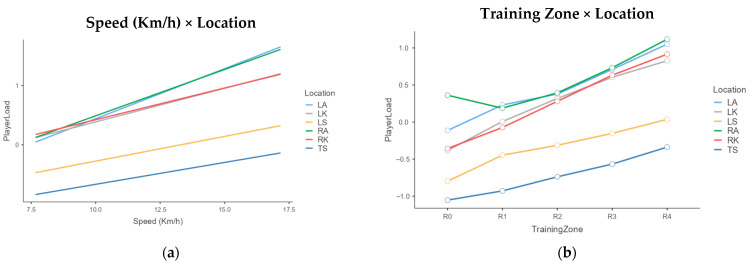
Interaction profiles for PlayerLoad as a function of location, speed, and training zone. (**a**) Relationship between Speed (Km/h) and Location. (**b**) Relationship between Training Zone and Location. LA: left ankle; RA: right ankle; RK: right knee; LK: left knee; LS: lumbar spine; TS: toracic spine; R0: 50–60% HRmax; R1: 60–70% HRmax; 70–80% HRmax; R3: 80–90% HRmax; R4: >90% HRmax.

**Table 1 sports-13-00268-t001:** Fixed effects omnibus tests for PlayerLoad in Model 1 and Model 2.

	F	df	df (res)	*p*	η*p*^2^
Model 1 (Location, Speed, and their Interaction)					
Location	3391.9	5	1607	**<0.001**	**0.912**
Speed (Km/h)	4555.5	1	1608	**<0.001**	**0.739**
Location × Speed	88.6	5	1607	**<0.001**	**0.216**
Model 2 (Location, Training Zone, and their Interaction)					
Location	497.5	5	1589	**<0.001**	**0.610**
Training Zone	744.2	4	1590	**<0.001**	**0.652**
Location × Training Zone	11.6	20	1589	**<0.001**	**0.127**

F values are rounded to two decimal places. df = degrees of freedom of the effect; df (res) = residual degrees of freedom; *p* = *p*-value. *p* < 0.001 indicates a *p*-value less than 0.001.

**Table 2 sports-13-00268-t002:** Fixed coefficients for PlayerLoad in Model 1.

Effect	Estimate	SE	IC 95%		df	t	*p*
			Lower	Upper			
Intercept	0.32	0.04	0.23	0.42	6.99	6.81	**<0.001**
Location							
LK-LA	−0.14	0.01	−0.17	−0.11	1.607.62	−10.63	**<0.001**
LS-LA	−0.85	0.01	−0.88	−0.83	1.606.99	−67.33	**<0.001**
RA-LA	0.02	0.01	0.06	0.05	1.607.62	2.01	**0.045**
RK-LA	−0.12	0.01	−0.14	−0.09	1.606.99	−9.42	**<0.001**
TS-LA	−1.26	0.01	−1.29	−1.24	1.606.99	−99.28	**<0.001**
Speed (Km/h)	0.11	0.01	0.11	0.12	1.608.17	67.49	**<0.001**
LK–LA × Speed	−0.05	0	−0.06	−0.04	1.607.05	−9.42	**<0.001**
LS-LA × Speed	−0.08	0	−0.09	−0.07	1.606.99	−15.18	**<0.001**
RA-LA × Speed	−0.01	0	−0.02	−0.01	1.607.00	−2.13	**0.003**
RK-LA × Speed	−0.06	0	−0.07	−0.05	1.606.99	−11.01	**<0.001**
TS-LA × Speed	−0.09	0	−0.10	−0.08	1.606.99	−16.89	**<0.001**

LA: left ankle; RA: right ankle; RK: right knee; LK: left knee; LS: lumbar spine; TS: toracic spine; Reference category for location: LA (left ankle). *p* values < 0.001 are indicated for values less than 0.001. Values are presented as estimate (standard error); 95% CI = 95% confidence interval.

**Table 3 sports-13-00268-t003:** Fixed coefficients for PlayerLoad in Model 2.

Effect	Estimate	SE	IC 95%		df	t	*p*
			Lower	Upper			
Intercept	0.08	0.07	−0.05	0.21	7.23	1.26	0.246
Location							
LK-LA	−0.02	0.03	−0.24	−0.11	1589.05	−5.32	<0.001
LS-LA	−0.08	0.03	−0.85	−0.72	1589.00	−23.94	<0.001
RA-LA	0.10	0.03	0.05	0.17	1589.09	3.51	<0.001
RK-LA	−0.02	0.03	−0.23	−0.11	1589.00	−5.23	<0.001
TS-LA	−1.17	0.03	−1.24	−1.11	1589.00	−35.82	<0.001
Training Zone							
R1-R0	0.02	0.04	0.13	0.30	1589.51	4.88	<0.001
R2-R0	0.44	0.04	0.35	0.53	1589.50	10.28	<0.001
R3-R0	0.72	0.04	0.63	0.80	1589.75	16.63	<0.001
R4-R0	0.99	0.04	0.90	1.07	1589.67	23.03	<0.001
Location × Training Zone							
LK-LA × R1-R0	0.04	0.15	−0.26	0.35	1589.00	0.29	0.765
LS-LA × R1-R0	0.01	0.15	−0.29	0.31	1589.00	0.05	0.959
RA-LA × R1-R0	−0.51	0.14	−0.79	−0.23	1589.02	−3.61	<0.001
RK-LA × R1-R0	−0.05	0.15	−0.36	0.24	1589.00	−0.38	0.702
TS-LA × R1-R0	−0.21	0.15	−0.52	0.08	1589.00	−1.39	0.162
LK-LA × R2-R0	0.21	0.15	−0.08	0.50	1589.00	1.38	0.166
LS-LA × R2-R0	−0.01	0.15	−0.30	0.28	1589.00	−0.04	0.965
RA-LA × R2-R0	−0.45	0.13	−0.72	−0.19	1589.02	−3.36	<0.001
RK-LA × R2-R0	0.14	0.15	−0.15	0.44	1589.00	0.96	0.336
TS-LA × R2-R0	−0.17	0.15	−0.47	0.11	1589.00	−1.17	0.241
LK-LA × R3-R0	0.16	0.14	−0.12	0.45	1589.00	1.09	0.273
LS-LA × R3-R0	−0.17	0.14	−0.47	0.11	1589.00	−1.18	0.235
RA-LA × R3-R0	−0.45	0.13	−0.71	−0.18	1589.03	−3.35	<0.001
RK-LA × R3-R0	0.16	0.14	−0.12	0.45	1589.00	1.11	0.267
TS-LA × R3-R0	−0.33	0.14	−0.63	−0.04	1589.00	−2.25	0.024
LK-LA × R4-R0	0.04	0.15	−0.24	0.34	1589.00	0.31	0.756
LS-LA × R4-R0	−0.32	0.14	−0.62	−0.03	1589.00	−2.19	0.028
RA-LA × R4-R0	−0.40	0.13	−0.67	−0.14	1589.03	−3.02	0.003
RK-LA × R4-R0	0.10	0.14	−0.18	0.40	1589.00	0.72	0.467
TS-LA × R4-R0	−0.44	0.14	−0.74	−0.15	1589.00	−2.99	0.003

LA: left ankle; RA: right ankle; LK: left knee; RK: right knee; LS: lumbar spine; TS: thoracic spine. Reference category for location: LA (left ankle). Reference category for training zone: R0 (50–60% HRmax). *p* values < 0.001 are indicated for values less than 0.001. Values are presented as estimate (standard error); 95% CI = 95% confidence interval.

## Data Availability

The data supporting the findings of this study are openly available in DIGITUM, the Institutional Repository of the University of Murcia, at http://hdl.handle.net/10201/156980 (accessed on 1 July 2025).

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
