# Peer review of "Segmental External Load in Linear Running in Elite Futsal Players: A Multifactorial and Individual Variability Analysis Using Linear Mixed Models"

_sports, 2025, doi:10.3390/sports13080268_

Round 1

Reviewer 1 Report

Comments and Suggestions for Authors

Dear authors,

Your manuscript is carefully structured and well-written. However, there is a critical issue concerning the statistical analysis. Below you will find the detailed evaluation of my review.

Abstract

The abstract is well formulated, with a clear presentation of the study aim, methodology, main findings, and practical implications.

Introduction

The introduction is clearly structured, with thorough justification of the research question, appropriate literature support, and logical flow toward the study objectives. The use of Linear Mixed Models is appropriately emphasized as an innovative choice for modeling multifactorial effects and individual variability.

Materials and Methods

Excellent and detailed work is evident in sections 2.1–2.2 and 2.5–2.6. However, in sections 2.3 and 2.4, which address the variables and statistical analysis, there is a critical problem: the variables Speed and Training Zone cannot be included in the same model due to near-perfect correlation. I ran a correlation analysis using data from one player and found r = 0.967, which is well above the commonly accepted thresholds (0.8 or even 0.9). I recommend either removing one of the two variables from the model or running two separate models instead. Suggested independent variables per model:

  1. Location, Speed, Location × Speed
  2. Location, Training Zone, Location × Training Zone

Remaining sections

These cannot be properly evaluated as they are based on results derived from flawed statistical modeling. I would only highlight two points:

  1. In the Results section, effect sizes should be reported, which are currently missing from this version;
  2.  When examining interactions, we do not run a separate model with only the interactions. Interactions should be tested within the same model that includes the main effects.

Reviewer 2 Report

Comments and Suggestions for Authors

This manuscript titled “Segmental External Load in Linear Running in Elite Futsal Players” addresses a timely and relevant topic in the field of performance monitoring and sports biomechanics. The use of inertial measurement units and linear mixed models to examine segmental PlayerLoad across anatomical regions and intensities in elite female futsal players is both innovative and methodologically appropriate.  A detailed list of specific recommendations by page (P) and line (L) is provided below to support further improvement of the manuscript.

Recommendations:

P1, L2–4: Clarify that this is an "observational, within-subject repeated measures study"

P2, L30–86: Add a final paragraph explicitly better the novelty of your study compared to Gómez-Carmona et al. and justify the use of elite female players - why this group matters (e.g., injury vulnerability, lack of data). 

P3, L88–100: Although the study was approved by the Ethics Committee and used anonymized data, high-impact journals often require written informed consent even for anonymized data. Consider rephrasing or ensuring this was in fact waived by the IRB (e.g., quote the committee waiver). Clarify inclusion/exclusion criteria beyond “no musculoskeletal limitations.”

P4, L111–134: Consider grouping variables in a table or figure (e.g., “Variable definitions and measurement procedures”) for clarity.

P4–5, L144–160: Include rationale for log-transformation of PlayerLoad and describe why this transformation improves model fit.

P7–8, L224–236: Emphasize effect sizes and confidence intervals, not only p-values, when interpreting meaningfulness.

P6–7: Consider including a table with descriptive stats (means ± SDs) for each segment across intensities/speeds to provide context beyond model coefficients.

P9–10, L254–326: The discussion would benefit from a clearer comparison with prior studies (e.g., does the knee overload replicate findings in other sports or populations?).

Discuss practical implications beyond training load (e.g., equipment design, injury prevention programs).

P10, L330–336: More robust discussion on why right knee overload may occur (e.g., leg dominance, asymmetry, motor patterns).

P11, L380–395: Be more precise. Suggest how the treadmill linearity differs from futsal constraints (e.g., cutting, pivoting), and how this may affect transferability.

P12, L420–422: Consider aligning the Data Availability Statement with FAIR principles, referencing metadata format and license type.

Round 2

Reviewer 1 Report

Comments and Suggestions for Authors

Dear authors,

I would like to sincerely congratulate you on the substantial improvements made in your revised manuscript. It is evident that you have carefully considered the comments and taken the necessary steps to enhance the quality and statistical rigor of your work.

I am particularly pleased with your decision to adopt two separate Linear Mixed Models (LMMs), as suggested. This approach effectively addresses the issue of multicollinearity between Speed and Training Zone. It is great to see how the revised models have not only improved the statistical power but also revealed clearer main effects and interactions, which add depth and precision to your findings.

Congratulations once again on the excellent work.